# Generalisable Agents for
# Neural Network Optimisation

**Kale-ab Tessera**[1*†]   **Callum Rhys Tilbury**[2*]   **Sasha Abramowitz**[2*]
**Ruan de Kock**[2]   **Omayma Mahjoub**[2]
**Benjamin Rosman**[3,4]   **Sara Hooker**[5]   **Arnu Pretorius**[2]

[1]University of Edinburgh   [2]InstaDeep Ltd   [3]The University of the Witwatersrand
[4]CIFAR Azrieli Global Scholar, CIFAR   [5]Cohere For AI

## Abstract

Optimising deep neural networks is a challenging task due to complex training dynamics, high computational requirements, and long training times. To address this difficulty, we propose the framework of Generalisable Agents for Neural Network Optimisation (GANNO)—a multi-agent reinforcement learning (MARL) approach that learns to improve neural network optimisation by dynamically and responsively scheduling hyperparameters during training. GANNO utilises an agent per layer that observes localised network dynamics and accordingly takes actions to adjust these dynamics at a layerwise level to collectively improve global performance. In this paper, we use GANNO to control the layerwise learning rate and show that the framework can yield useful and responsive schedules that are competitive with handcrafted heuristics. Furthermore, GANNO is shown to perform robustly across a wide variety of unseen initial conditions, and can successfully generalise to harder problems than it was trained on. Our work presents an overview of the opportunities that this paradigm offers for training neural networks, along with key challenges that remain to be overcome.

## 1   Introduction

Deep neural networks have optimisation landscapes that are non-convex and high-dimensional, resulting in complicated training dynamics (Li et al., 2018). Furthermore, due to their sheer size, training modern deep learning models is a particularly expensive endeavour that requires significant computational resources and time (Kaddour et al., 2023).

The issue of hyperparameters in training is of particular interest, due to both their significant influence on model performance and training speed (Schmidt et al., 2021), and the prohibitive computational costs of hyperparameter tuning (Sharir et al., 2020). Furthermore, once a value or schedule has been obtained, the result is often problem-specific, i.e. a set of parameters that is ideal for one model might not generalise to other problems differing in architecture or dataset.

Existing strategies for choosing hyperparameters struggle to simultaneously satisfy the requirements of performance, efficiency, and generalisability. Methods like grid-search and Bayesian optimisation (Feurer and Hutter, 2019), though straightforward, are tuned to a particular problem and are unlikely to generalise. Expert-derived heuristics, such as Google's Deep Learning Playbook (Godbole

---

[*]Equal contribution.

[†]Corresponding author: kaleabtessera[at]gmail[dot]com. Work done while a research engineer at InstaDeep Ltd.

Workshop on Advancing Neural Network Training at 37th Conference on Neural Information Processing Systems (WANT@NeurIPS 2023).

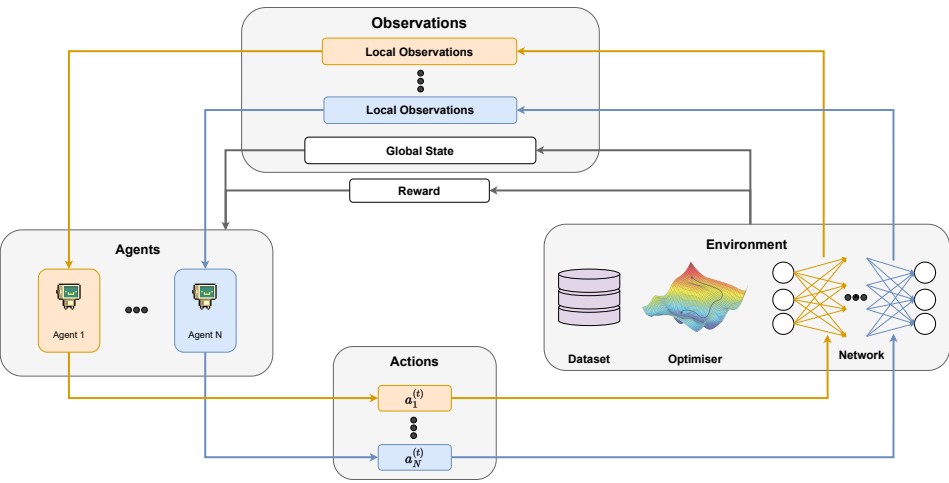

Figure 1: *GANNO's training process.* There is an agent per layer of a neural network. Each agent receives a set of global and layer-specific observations about the environment and uses this information to select an action, which is applied to a corresponding layer. Then, training in the environment progresses for some time, after which a reward signal is returned and this loop continues.

et al., 2023), are also often problem-specific, requiring reconsideration with each new context. Certain newer methods are instead data-driven—i.e., they leverage information from observed neural network dynamics to learn optimisation strategies, with the intention to generalise beyond their trained setting. One instance of this approach is to train an entirely new optimiser by meta-learning the weight-update rules for optimisation, based on the performance over a distribution of tasks (e.g. (Andrychowicz et al., 2016; Li and Malik, 2016; Wichrowska et al., 2017; Chen et al., 2022; Metz et al., 2022)). Though performant and generalisable, these methods carry a significant computational burden. Consider, for example, the VeLO optimiser (Metz et al., 2022), which required four-thousand TPU-months to train. Though this cost is arguably 'once-off' after training is complete, developing subsequent versions of this optimiser (e.g. for tasks unseen in the meta-training distribution, where Metz et al. (2022) acknowledge that it struggles) remains prohibitively expensive, which constrains this approach for future development of new optimisers for new problems.

A more compute-efficient approach, which remains data-driven, is to instead learn an optimisation schedule, rather than the optimiser itself. That is, employ an existing optimisation algorithm (e.g., SGD (Robbins and Monro, 1951)), but learn how to evolve its hyperparameters over time. Scheduling has widely been acknowledged for its potential to provide significant improvements in performance, especially when applied to the learning rate (Bottou, 2012; Sun et al., 2023), and it has been shown that it is possible to learn such schedules using reinforcement learning (RL). However, previous works using RL (e.g. (Xu et al., 2019; Almeida et al., 2021)) have taken a single-agent approach, and thus were constrained to learning an identical learning rate schedule for all layers, based on global network information, such as the training loss. Elsewhere though, it has been shown that setting layerwise learning rates is valuable (You et al., 2017, 2019), and therefore, this global constraint naturally limits the overall effectiveness of learning dynamic schedules for deep networks.

In this work, we build on the success of RL as a sequential decision-making paradigm for optimisation. We use the knowledge that operating at a layerwise level is useful, while avoiding problem-specific heuristics and remaining relatively computationally friendly. Hence, we propose Generalisable Agents for Neural Network Optimisation (GANNO): a novel, multi-agent reinforcement learning (MARL) approach to optimisation. GANNO leverages layerwise information to learn adaptive layerwise learning rate schedules, as depicted in Figure 1. We show that GANNO can learn competitive schedules when compared to other leading approaches and demonstrates robustness across a wide range of unseen initial conditions. Importantly, this robustness removes the need to know the optimal values for these parameters *a priori*. We further demonstrate generalisation, where GANNO can be used successfully in problems that are more complex than what it was trained on. Finally, we outline the core challenges in this paradigm and some avenues for future work.

## 2 Background

**Neural network optimisation.** We consider a neural network $f_{\boldsymbol{\theta}}$, parameterised by learnable weights $\boldsymbol{\theta}$. Given a training dataset $D = \left\{ \left(x^{(m)}, y^{(m)}\right) \right\}_m^M$ containing $M$ examples, we aim to minimise the objective, $J(\boldsymbol{\theta}; \lambda) = \mathbb{E}_{\boldsymbol{x}, y \sim \hat{p}_{\text{data}}(\boldsymbol{x}, y)}[L(f(\boldsymbol{x}; \boldsymbol{\theta}), y; \lambda)]$, where $L$ is a loss function evaluated using the predictions from the model $f(\boldsymbol{x}^{(m)}; \boldsymbol{\theta})$ and the true labels from the dataset $y^{(m)}$. We notate $\hat{p}_{\text{data}}$ as the empirical distribution over the training set, and $\lambda$ as the weight decay coefficient.

To minimise this objective, we consider optimisation methods that adopt an update rule of the form, $\boldsymbol{\theta}^{(\tau+1)} \leftarrow \boldsymbol{\theta}^{(\tau)} - \phi\left(\nabla_{\boldsymbol{\theta}} J(\boldsymbol{\theta}^{(\tau)}; \lambda^{(\tau)}), \boldsymbol{\theta}^{(\tau)}; \alpha^{(\tau)}\right)$, where $\boldsymbol{\theta}^{(\tau)}$ are the current parameters at step $\tau$ and $\boldsymbol{\theta}^{(\tau+1)}$ are the updated parameters. $\phi$ is the chosen optimiser (e.g. Adam (Kingma and Ba, 2015)) and is parameterised by $\alpha^{(\tau)}$ (e.g. the learning rate) and is a function of the gradient of the loss with respect to the parameters, $\nabla_{\boldsymbol{\theta}} J(\boldsymbol{\theta}^{(\tau)}; \lambda^{(\tau)})$ and the parameters themselves, $\boldsymbol{\theta}^{(\tau)}$.

**Multi-agent reinforcement learning (MARL).** We consider the case of common-reward cooperative MARL, which can be formulated as a decentralised partially-observable Markov decision process (Bernstein et al., 2002) with a set of $N$ agents, $\mathcal{N} = \{1, \ldots, N\}$, a state space $\mathcal{S}$, a joint-observation space $\mathcal{O} = (\mathcal{O}_1 \times \cdots \times \mathcal{O}_N) \subseteq \mathcal{S}$, and a joint-action space $\mathcal{A} = \mathcal{A}_1 \times \cdots \times \mathcal{A}_N$. At each discrete timestep $t$, the agents exist in a state $s^{(t)} \in \mathcal{S}$, where each agent $i$ perceives its own observation $o_i^{(t)} \in \mathcal{O}_i$ and accordingly takes its own action $a_i^{(t)} \in \mathcal{A}_i$. Based on the joint action, the agents transition to a next state $s^{(t+1)} \in \mathcal{S}$, with probabilities defined by a transition distribution $\mathcal{P} : \mathcal{S} \times \mathcal{A} \times \mathcal{S} \rightarrow [0, 1]$, and receive a shared scalar reward, $r^{(t)}$ from the reward function $\mathcal{R} : \mathcal{S} \times \mathcal{A} \times \mathcal{S} \rightarrow \mathbb{R}$. The agents' return is defined by their discounted cumulative rewards, $G = \sum_t^T \gamma^t r^{(t)}$, where $T$ is the number of time steps in an episode, and $\gamma \in (0, 1]$ is a discounting factor. Each agent's policy is given by $\pi_i(a_i | o_i)$, with the set of all agents' policies as $\pi = \{\pi_1, \ldots, \pi_N\}$. The objective in cooperative MARL is to find a policy $\pi_i$ for each agent $i$ such that the return is maximised with respect to the other agents' policies, $\pi_{-i} := \{\pi \backslash \pi_i\}$. That is, $\forall i : \pi_i \in \arg\max_{\hat{\pi}_i} \mathbb{E}\left[G | \hat{\pi}_i, \pi_{-i}\right]$.

**Notions of generalisation in RL.** Along with Metz et al. (2020) and Almeida et al. (2021), we assert that generalisation is a critical component of learned optimisers. Yet generalisation, particularly in the context of reinforcement learning, can often lack a consistent definition. Here, we adopt an environment categorisation introduced by Kirk et al. (2023), where environments may be (1) singleton, (2) independent and identically distributed (IID), or (3) out-of-distribution (OOD). When learning optimisers with this categorisation, environment generalisation can occur across various axes of the environment components, such as generalisation across various combinations of $f_{\boldsymbol{\theta}}$, $D$, $L$, and $\phi$. In this work, we focus on IID generalisation of the neural network $f_{\boldsymbol{\theta}}$ and OOD generalisation of the dataset $D$. We consider it important to keep the specific area of generalisation clear and encourage future work into more complex levels of generalisation to do so as well.

## 3 Related Work

Our work primarily relates to three key insights from the literature on neural network optimisation.

**Scheduling is useful.** Using a schedule for hyperparameters has been recommended in training neural networks for several decades (Darken et al., 1992), and many subsequent works have sought to find good schedules via a wide variety of strategies (Loshchilov and Hutter, 2016; Smith, 2017; Smith and Topin, 2019). Importantly, many of these scheduling approaches are based on simple heuristics, developed using the observations of practitioners with experience in the field (e.g. (Godbole et al., 2023)). We aim to employ the power of scheduling in our work, but in a dynamic and responsive way, avoiding the need for handcrafted functions.

**Layerwise information is important.** The Layerwise Adaptive Rate Scaling (LARS) method (You et al., 2017) adapts stochastic gradient descent (SGD) to have layerwise learning rates, where a defined global rate is scaled for each layer by the ratio between the norm of that layer's weights and the norm of the gradient updates, referred to as the trust ratio. LAMB (You et al., 2019) extends this approach to use Adam (Kingma and Ba, 2015) and additionally considers weight decay. These techniques demonstrate faster convergence times showing that layerwise information is useful. However, the trust

ratio fundamentally remains a handcrafted heuristic, which happens to work well in certain domains and not necessarily in others (You et al., 2019). We aim to leverage the layerwise information in a network while avoiding handcrafted heuristics, with the end goal of generalisation.

**RL is effective for learning schedules.** Most closely related to our work are methods which learn data-driven optimisation schedules using RL (Xu et al., 2019; Almeida et al., 2021; Xiong et al., 2022). These works highlight the potential usefulness of such a strategy; however, none of them operate in a layerwise manner—considering layer-specific dynamics and taking layer-specific actions. Thus, we aim to extend these RL approaches to a multi-agent setting using a separate agent per layer.

## 4    Methodology

In this section, we present GANNO: Generalisable Agents for Neural Network Optimisation. GANNO is a general framework that uses MARL to train agents to observe aspects of a neural network $f_\theta$ during supervised learning, and develop a policy for selecting the optimiser hyperparameters dynamically during the learning process. Figure 1 provides a high-level illustration of how GANNO works: each layer passes observations to its corresponding agent; agents make decisions on how to adjust the hyperparameters; the neural network $f_\theta$ is trained for $\tau$ steps; and a reward is yielded from the performance of supervised learning. Note that in this work, we constrain the parameters under control, $\alpha$, to be adjustments of the learning rate; however, the framework can be extended to other hyperparameters of interest—e.g., future work could explore using GANNO to control both the learning rate and weight decay parameters simultaneously. We describe below the details of our MARL formulation.

**Timescale.**  Each timestep $t$ in the MARL environment corresponds to $\tau$ steps of training in the underlying neural network, $f_\theta$. Acting too frequently (e.g. $\tau = 1$) makes the environment highly non-stationary, making it difficult for agents to learn the impact of their actions. On the other hand, acting too infrequently slows down training. Empirically, we find that acting with $\tau = 100$ (i.e. every 100 gradient updates of $f_\theta$) performs well.

**Environments.**  We make an important delineation between train and evaluation environments. The former is a neural network with a particular architecture, dataset, and optimiser, which is used when our MARL agents are training. The latter, in contrast, is only run *after* the MARL agents have been trained, and in this case, the neural network can be the same or different to the one used as the training environment. We are particularly interested in evaluating generalisation, which is the case when the evaluation environment differs from the training environment and is more complex.

**Observations.**  Each agent receives a shared global observation along with a set of local observations specific to that agent's own layer. An example of a global observation is the current training loss, as this metric is the same across all layers. In contrast, a local observation could be the mean of the neural network weights of a layer. Details of all observations can be found in Appendix A.

**Actions.**    Our agents operate in a discrete action space, taking actions to modify the current learning rate. Each action consists of a mathematical operation $\oplus$ with a corresponding value $x$. The modification to the learning rate is then $\alpha \oplus x$. For example, with the action $\{\oplus = +; x = 0.001\}$, the agent adds 0.001 to the current learning rate.

**Reward.**  For the reward, we use the classification accuracy on a hold-out validation dataset used in the training environment,[3] to encourage the learning of generalisable behaviour. Importantly, when using solely this metric as a reward, our agents cannot tell if their actions directly resulted in a change in reward, or if the neural network's performance simply changed as a result of progress in the training of $f_\theta$. To handle this problem, we leverage difference rewards (Wolpert and Tumer, 2001; Proper and Tumer, 2012), where instead of using a reward signal $r^{(t)} = \mathcal{R}(s^{(t)}, a^{(t)}, s^{(t+1)})$, we shape the reward $r^{(t)} = \mathcal{R}(s^{(t)}, a^{(t)}, s^{(t+1)}) - \mathcal{R}(s^{(t)}, \tilde{a}, s^{(t+1)})$, where $\tilde{a}$ is the action of no modification to the current learning rate (often referred to as a 'no-op' action). Note that this modification requires two steps of training the neural network $f_\theta$ (one with each action, $a$ and $\tilde{a}$) at each MARL timestep $t$, but this extra step only occurs during training and is not necessary during evaluation.

---

[3]Note that since we aim to generalise to unseen datasets during evaluation, there is no chance of dataset contamination.

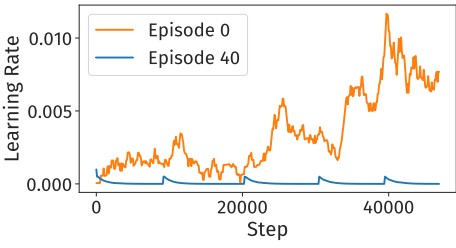
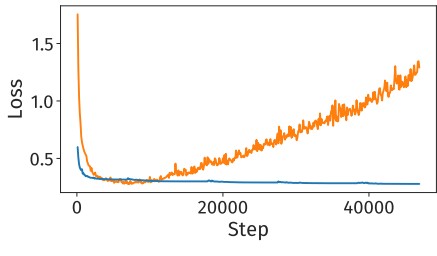

(a) Learning rate schedule.      (b) Classification loss.

Figure 2: *GANNO's dynamic learning rate and corresponding training loss on* `Fashion-MNIST`, *shown at episodes 0 and 40.* The first episode of MARL training is shown in orange and in later training, at episode 40, in blue. Both training and evaluation use a two-layered CNN. We observe clear evidence of a useful schedule being learned, which improves the classification loss.

**Initial conditions.** We aim for GANNO to have competitive performance across a wide range of initial conditions. Accordingly, to encourage such generalisation across starting conditions, we sample different values for the initial learning rate $\alpha_{\text{init}}$ and weight decay $\lambda$ used in the training environment. Specifically, we use a log-uniform distribution to yield samples uniformly across different orders of magnitude. In evaluation, we use a fixed set of reasonable initial values for the problem, to robustly assess the performance of different approaches.

**Agent policies.** We use independent proximal policy optimisation (IPPO) (Schulman et al., 2017; de Witt et al., 2020)[4], where each agent's policy is parameterised by a recurrent neural network, with parameters $\xi_i$. We use weight sharing for efficient training by setting $\xi = \xi_1 = \cdots = \xi_N$. We still enable agent specialisation by conditioning each agent's policy on information local to that agent as well as an embedding of the agent's depth in the network. Note that this shared-parameter formulation naturally enables depth generalisation: we can train on a network with $L_1$ layers, yet evaluate on a network with $L_2 > L_1$ layers, while avoiding an observation-dimension mismatch due to having more layers in evaluation.

## 5 Results

We perform several experiments to validate our approach. We find that GANNO produces useful schedules which are responsive and robust, while being capable of generalising to more difficult problems. We further show that our MARL formulation is crucial for such capabilities. We use the Adam (Kingma and Ba, 2015) optimiser and unless otherwise stated, the hyperparameters used are those listed in Appendix A. Values in tables are given with one standard deviation over three seeds, and boldface indicates the largest value in a column.

**Useful and responsive schedules are generated.** We first consider the simplest case of GANNO, without any notion of generalisation. Here we train and evaluate our MARL system on identical environments—the same network architecture, optimiser, and dataset. We use a two-layered convolutional neural network (CNN), applied to `Fashion-MNIST` (Xiao et al., 2017). Figure 2 shows two instances of the learning rate that GANNO yields in this setting, at episode zero at the beginning of training, and then later in training at episode 40, along with their corresponding loss curves.

We highlight several interesting insights. Firstly, we observe clear learning taking place. GANNO outputs a random learning rate schedule during the first episode, which results in an undesirable loss curve; yet later in training, it yields a much improved dynamic schedule, resulting in a more desirable loss curve. Secondly, this dynamic schedule is reminiscent of some of the leading handcrafted schedules in the literature. We see similarity to exponential decay in the early stages of learning, and cyclical patterns akin to SGDR (Loshchilov and Hutter, 2016) as training progresses.

Lastly, GANNO seems to instil in the learning rate schedule the ability to escape local optima in the loss landscape dynamically during training. To demonstrate this, we plot Figure 3, which shows another instance of training and evaluating on `Fashion-MNIST` with a two-layered CNN. We show

---

[4]Implemented using Mava (Pretorius et al., 2021), a MARL framework.

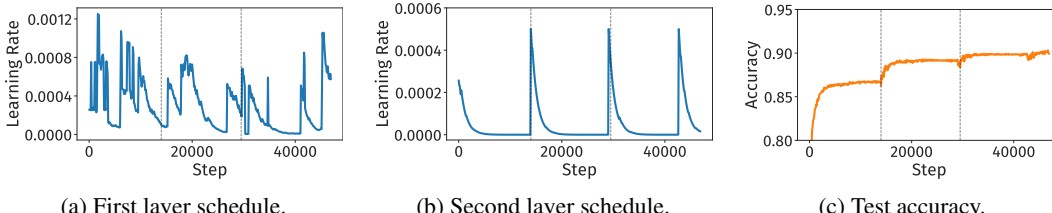

| (a) First layer schedule. | (b) Second layer schedule. | (c) Test accuracy. |

Figure 3: *GANNO's learning rate schedule dynamically escaping local optima during training for a two-layer CNN on* `Fashion-MNIST`. *In both layers at key moments (around* $14\,000$ *and* $29\,000$ *training steps), GANNO spikes the learning rate and thereby escapes a local optima and improves performance.*

the two layers' learning rate curves and the corresponding test accuracy. Here, the learned strategy for scheduling, particularly in Layer 1, is not as refined as seen in Figure 2. However, we draw attention to the two significant jumps in training accuracy at around $14\,000$ and $29\,000$ training steps—corresponding to spikes in the learning rate: firstly in Layer 2, and then in Layer 1. At this point in training, GANNO shows clear evidence of helping the neural network escape local optima, improving the test accuracy by several percentage points each time. Moreover, we observe how the layerwise learning rates coordinate to achieve this. This observation demonstrates the power of a responsive, layerwise scheduling algorithm. We see how GANNO can do more than simply yield a schedule akin to the handcrafted schedules from the literature, by acting dynamically based on the the layerwise information it observes.

**Signs of generalisation and robustness.** An important aim of GANNO is to generalise to problems of different levels of complexity. For example, to train on a simpler, shallower neural network, and still capture the dynamics well enough to be evaluated zero-shot on a more complex, deeper network. To investigate GANNO's ability to generalise in this way, we experiment by training on a two-layered CNN as before, but now evaluating on a five-layered CNN. Furthermore, we train on `Fashion-MNIST`, but evaluate on `CIFAR-10` (Krizhevsky et al., 2009), with the latter being a more complex dataset. We compare these methods across a range of initial learning rates during evaluation, with the results given in Table 1. To benchmark our performance, we include the results of using various manual learning rate baselines from the literature, initialised across the same learning rate values, as well as two meta-learned optimisers, VeLO (Metz et al., 2022) and Lion (Chen et al., 2023).

When comparing GANNO to manual schedules, we see that although GANNO is not the best-performing schedule, it performs well across initial learning rate conditions, thus indicating robustness, while remaining competitive with popular expertly handcrafted schedules. In Appendix B, we show

Table 1: *Test accuracy (%) when generalising to a five-layered CNN on* `CIFAR-10`, *using manual, learned schedules and learned optimisers.* We compare GANNO to several manual learning rate schedules and two learned optimisers. This comparison is done across various initial learning rates, except for VeLO which does not take a learning rate parameter. All experiments are done with a weight decay of $\lambda = 0.1$, with additional experiments using $\lambda = 0.01$ for VeLO and Lion since they failed to generalise when using $\lambda = 0.1$. We find that GANNO performs competitively compared to the baselines and on average, is the third most performant approach across initial learning rates. We find this to be consistent across smaller $\lambda$ values as shown in Table 5 in Appendix C.

| | Initial learning rate | | | | | |
|---|---|---|---|---|---|---|
| **Method** | 0.0001 | 0.0003 | 0.001 | 0.003 | 0.01 | Average |
| Constant | $71.99 \pm 0.38$ | $71.33 \pm 0.20$ | $71.95 \pm 0.43$ | $73.25 \pm 0.73$ | $62.34 \pm 0.45$ | $70.17 \pm 0.21$ |
| Linear decay | $71.27 \pm 0.58$ | $72.51 \pm 0.52$ | $72.77 \pm 0.13$ | $73.19 \pm 0.39$ | $69.87 \pm 0.72$ | $71.92 \pm 0.23$ |
| Exponential decay | $69.67 \pm 0.59$ | $72.99 \pm 0.52$ | $72.55 \pm 0.24$ | $72.47 \pm 0.54$ | $68.97 \pm 0.74$ | $71.33 \pm 0.25$ |
| SGDR (Loshchilov and Hutter, 2016) | $70.72 \pm 0.28$ | $72.90 \pm 0.36$ | $73.79 \pm 0.50$ | $\mathbf{74.83 \pm 0.09}$ | $71.36 \pm 2.44$ | $72.72 \pm 0.51$ |
| LAMB (You et al., 2019) w/ cosine decay | $62.43 \pm 0.38$ | $69.65 \pm 0.18$ | $71.5 \pm 0.15$ | $72.58 \pm 0.13$ | $\mathbf{75.48 \pm 0.33}$ | $70.33 \pm 0.11$ |
| VeLO (Metz et al., 2022), $\lambda = 0.1$ | / | / | / | / | / | $10.00 \pm 0.00$ |
| Lion (Chen et al., 2023), $\lambda = 0.1$ | $71.53 \pm 0.28$ | $\mathbf{73.82 \pm 0.14}$ | $55.34 \pm 17.27$ | $10.00 \pm 0.00$ | $10.00 \pm 0.00$ | $44.14 \pm 3.54$ |
| VeLO (Metz et al., 2022), $\lambda = 0.01$ | / | / | / | / | / | $\mathbf{76.16 \pm 0.25}$ |
| Lion (Chen et al., 2023), $\lambda = 0.01$ | $71.46 \pm 0.13$ | $73.25 \pm 0.12$ | $44.90 \pm 8.44$ | $10.00 \pm 0.00$ | $10.00 \pm 0.00$ | $41.96 \pm 1.74$ |
| GANNO | $\mathbf{72.93 \pm 0.21}$ | $72.88 \pm 0.80$ | $\mathbf{74.32 \pm 0.16}$ | $73.79 \pm 0.26$ | $67.12 \pm 2.05$ | $72.21 \pm 0.70$ |

depictions of various manual schedules in Figure 5 and provide a full set of results for these schedules in Table 4. These results also highlight that expert-derived learning rate schedules, notably SGDR, are competitive baselines.

**Competing with learned optimisers.** We also compare GANNO to VeLO (Metz et al., 2022) and Lion (Chen et al., 2023). We see in Table 1 that while GANNO performs better than Lion, it remains worse than VeLO on this benchmark. VeLO's impressive performance indicates that it has learned useful parameter update rules distinctively different from Adam. Even with the promise of VeLO, it has some challenges. Notably, we see that it performs poorly using a weight decay of $\lambda = 0.1$, hinting that it is sensitive to $\lambda$ values. This could be problematic in compute-intensive tasks since sensitive values of $\lambda$ are often unknown before evaluating on a task. Furthermore, meta-learned optimisers like VeLO require exceptionally more compute to train. We discuss this in more detail in Section 6.

**Generalising to deeper networks.** We now consider GANNO's generalisation ability in a more complex setting: training on a residual network (He et al., 2016) that is 9 layers deep (ResNet-9) on `Fashion-MNIST`, and evaluating on one that is 18 layers deep (ResNet-18) on `CIFAR-10`. We compare GANNO to VeLO, the best-performing method from our smaller-scale experiments, along with simply using random layerwise agents. This comparison is done across two weight decay values, $\lambda = \{0.01, 0.1\}$. These results are shown in Figure 4 (we include results for SGDR, the best-performing manual schedule, in Figure 7 in Appendix C).

We see that GANNO performs competitively with VeLO. Furthermore, we find that it is more robust across the weight decay conditions compared to VeLO—which struggles to learn with a higher weight decay value, which is consistent with the results presented on a five-layer CNN. Moreover, the poor performance of the random agent demonstrates the difficulty of this problem (i.e. learning a responsive learning rate schedule for 18 layers), and thus the significance of GANNO's performance.

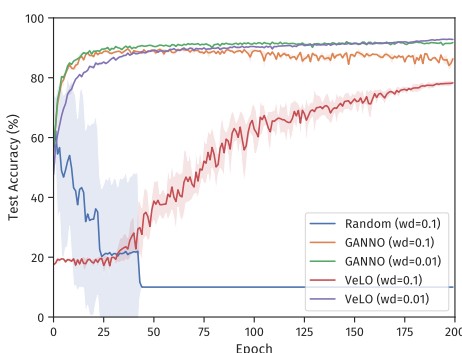

Figure 4: *Robustness of GANNO on ResNet-18.* Test accuracy across epochs for a random agent, VeLO and GANNO, evaluated on ResNet-18 on `CIFAR-10`, with an initial learning rate of $0.001$ for GANNO and the random agent. We see that GANNO produces robust and competitive schedules better able to handle different weight decay values.

It is promising that our framework can successfully generalise and control the hyperparameters of a network with vastly different dynamics than it was trained on. We see this evidence as a signal that GANNO is able to generalise to harder problem contexts (different layer depth, dataset difficulty), with robustness across unseen starting states (initial learning rate and weight decay values).

**The necessity of MARL.** In the previous results, we see early signs that GANNO is able to generate useful schedules and generalise to harder problem contexts. We now consider two further questions: is GANNO actually making use of the neural network dynamics to develop its control strategy? And is it important to observe such dynamics at a layerwise level? Accordingly, we study three ablated versions of GANNO: (1) `GANNO-LR-only`, where only the current learning rate is included in the observation, (2) `GANNO-timestep-only`, where only the training progress (current epoch count / total epoch count) is included in the observation, and (3) `GANNO-single-agent`, a single-agent version of GANNO where only global information (e.g. classification loss) is included in the observation and the agent learns a global learning rate schedule.[5] Table 2 shows the outcome of these experiments, with the same simplified problem configuration as used previously: training on a two-layered CNN with `Fashion-MNIST` and evaluating on a five-layered CNN with `CIFAR-10`.

Firstly, we notice that `GANNO-LR-only` performs well with some of the initial learning rates, specifically $0.003$, but deteriorates at other values. Notably, with an initial learning rate of $0$, it achieves a poor accuracy of just 35%. Studying the schedules that `GANNO-LR-only` yields, we find that agents

---

[5]Note this version would be comparable to work by Xu et al. (2019); Almeida et al. (2021). We were unable to find working code implementations for these methods so we implemented our own single-agent PPO agent with the same hyperparameters as our GANNO MARL agent.

Table 2: *An ablation study showing the necessity of GANNO's MARL formulation for learning dynamic schedules.* We show classification accuracies (%) using a five-layered CNN on `CIFAR-10` achieved by GANNO, along with the three ablations, all trained with a two-layered CNN on `Fashion-MNIST`, across various initial learning rates. We see that our GANNO formulation performs better than the ablated iterations, showing the necessity of observing layerwise dynamics and taking layerwise actions.

| | Initial learning rate | | | | | | |
|---|---|---|---|---|---|---|---|
| **Ablations** | 0 | 0.0001 | 0.0003 | 0.001 | 0.003 | 0.01 | Average |
| GANNO-LR-only | 35.80 ± 0.69 | 62.85 ± 0.55 | 70.13 ± 0.30 | 64.64 ± 0.91 | 74.27 ± 0.65 | 63.48 ± 2.83 | 61.54 ± 0.65 |
| GANNO-timestep-only | 57.81 ± 15.29 | 68.99 ± 4.51 | 73.04 ± 2.11 | 72.37 ± 3.88 | **74.39 ± 0.97** | 63.69 ± 2.48 | 68.38 ± 6.84 |
| GANNO-single-agent | 9.74 ± 0.11 | 51.32 ± 0.46 | 60.37 ± 0.74 | 69.24 ± 0.30 | 73.63 ± 0.37 | 64.10 ± 1.29 | 54.73 ± 0.54 |
| GANNO | **69.38 ± 1.47** | **72.93 ± 0.21** | **72.88 ± 0.80** | **74.32 ± 0.16** | 73.79 ± 0.26 | **67.12 ± 2.05** | **71.74 ± 0.83** |

often learn to simply decay the learning rate directly to zero, irrespective of the impact of this on the network dynamics. With an appropriate initial value, this strategy actually works reasonably well; but it is evidently not generalisable whatsoever, since it is not truly adaptive, leading to poor results across initial conditions.

For `GANNO-timestep-only`, we see that the training stage is indeed a useful observation, achieving relatively good performance across a fairly wide range of initial conditions. Nonetheless, we see again that this version of GANNO underperforms compared to the original, thus motivating the inclusion of network dynamics in our observations. Moreover, we witness much higher variance in the performance of `GANNO-timestep-only`, making the approach less reliable and robust.

Finally, we find that `GANNO-single-agent` significantly underperforms the layerwise version, and also fails to learn a generalisable schedule across initial conditions. This outcome clearly supports the usefulness of observing dynamics at a layerwise granularity and layerwise learning scheduling, as suggested in previous work (You et al., 2017, 2019).

## 6    Challenges, Opportunities, and Future Work

We believe our results indicate that the GANNO formulation for controlling network dynamics is a powerful one, which opens up promising research directions. Nonetheless, there remain several key challenges which we identify in this section. We specifically enumerate three primary dimensions: (1) agent foresight, (2) understanding agent success, and (3) computational requirements.

**Agent foresight is necessary for great performance.**  It is both common and useful in supervised learning to 'warm up' the learning rate hyperparameter during training—that is, use lower values when starting training, increase them in some way, and thereafter proceed with a scheduling strategy like exponential decay (He et al., 2016; Goyal et al., 2017; Godbole et al., 2023). In Table 3, we show the results of two such approaches when evaluating with a five-layered CNN on `CIFAR-10`: a simple strategy with linear warm-up and cosine decay, and the 'cosine one-cycle' schedule (Smith and Topin, 2019). We compare these schedules to GANNO's performance in this evaluation environment, setting its initial learning rate to zero to induce warm-up behaviour, after training it with a two-layered CNN on `Fashion-MNIST`.

We see in these results that the warm-up schedules, particularly with a good peak learning rate selection, are the most performant, achieving up to 77% in this classification task—the best results on

Table 3: *GANNO compared to warm-up schedules.*  We show classification accuracies (%) achieved using a five-layered CNN on `CIFAR-10` by GANNO, trained with a two-layered CNN on `Fashion-MNIST`, along with two leading warm-up schedules, across various peak learning rates. We see that the warm-up schedules achieve higher accuracies than GANNO.

| | Peak learning rate | | | | | |
|---|---|---|---|---|---|---|
| **Warm-up schedules** | 0.0001 | 0.0003 | 0.001 | 0.003 | 0.01 | Average |
| Linear warm-up, cosine decay | 70.77 ± 0.45 | 72.36 ± 0.45 | **73.61 ± 0.06** | **75.84 ± 0.26** | 77.24 ± 0.32 | 73.96 ± 0.15 |
| Cosine one-cycle (Smith and Topin, 2019) | **70.80 ± 0.16** | **72.71 ± 0.51** | 73.29 ± 0.11 | 75.80 ± 0.35 | **77.62 ± 0.60** | **74.04 ± 0.18** |
| GANNO from LR = 0 | / | / | / | / | / | 69.38 ± 1.47 |

this evaluation environment in this paper. In contrast, when we evaluate GANNO using an initial learning rate of zero, we see inferior performance.

The challenge here rests in the tricky balance between venturing into high learning rate regions while maintaining learning stability. We know from Table 1 that a constant learning rate at a high value performs poorly, yet we now observe in Table 3 that a great strategy is to increase up to this high value and thereafter decrease it. Notice that for an agent to replicate this effective schedule, it must have the foresight to move into a potentially unstable state of learning, but only do so temporarily. Though recurrent policies can help with the longer-term planning required here (Hausknecht and Stone, 2015), we find that agents tend to be more conservative to avoid this potential instability altogether (see Figure 6 for an example of such behaviour). A promising direction for this problem is to use existing manual schedules as demonstrations: e.g. to generate offline data from the successful handcrafted routines, and use this data in an offline MARL pre-training step (Formanek et al., 2023), thereby showing the agents the benefits of warm-up-like schedules.

**Understanding agent success.** The reward signal is a vital component of reinforcement learning, though one which is often considered as a given, simply as a part of the environment definition. Yet designing a reward signal for a particular goal may be an important task in itself (Eschmann, 2021). Consider the challenge of defining a meaningful reward signal when the underlying environment is itself a supervised learning problem. Ultimately, we want to optimise some final metric, e.g. maximise classification accuracy. Thus, suppose we used the training accuracy of the supervised loop as our reward; we are faced with the question discussed earlier in this paper: is our agent receiving a 'good' reward because of its own 'good' actions, or simply because of progress in the underlying training loop? Indeed, to illustrate this point empirically, notice that the agent could yield a constant learning rate (by taking 'no-op' actions, leaving the value unchanged), and in Table 1, we see that such a schedule yields a decent performance of around 70%. Instead, we want our agents to find a schedule that can squeeze out the extra performance—e.g., reach scores of 77%, as seen in Table 3.

Various directions for future work could extend from this point, such as reward shaping (as was done in the 'difference' rewards, discussed earlier) or using a centralised critic to improve multi-agent credit assignment (Yu et al., 2022).

**Computational Requirements.** In RL, it often takes millions of timesteps to train effective agents (Mnih et al., 2013; Schulman et al., 2017). In the case of GANNO, we train our agents for 50 000 timesteps. This shorter timespan is a result of two considerations. Firstly, for each training step, we require $\tau$ (e.g. 100) gradient updates from a supervised learning setting, which makes each environment step relatively slow compared to other RL environments. Secondly, we want our approach to be viable without access to large compute. In contrast, methods like VeLO (Metz et al., 2022) are trained using a computational budget on the order of thousands of TPU months. This immense computational requirement, despite being 'once-off' after training is complete, makes subsequent development of similar methods impossible for most of the machine learning community. GANNO's formulation is comparably much cheaper, with the above results yielded in just under six hours on a single NVIDIA A100 GPU. This lower barrier to entry enables greater access, and thus more development opportunities.

# 7   Conclusion

We introduced GANNO, a MARL approach that is used to control the training of a neural network. We described the salient details of our solution—the observations, actions, and rewards. We then enumerated the strengths of our proposed framework, supported by empirical results: that responsive and robust schedules can be generated; that the framework demonstrates signs of generalisation ability, where we can perform well on environments more complex than we trained on; and that observing layerwise neural dynamics is important, thus validating our choice of utilising MARL. We also presented the core challenges and opportunities for this framework to flourish: in particular, the need for agent foresight, a good reward signal and computational challenges. In sum, this paper offers a novel paradigm for tackling neural network optimisation—one which demonstrates strong signs of viability. However, challenges remain, and with them, many avenues for exciting future work.

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

# A   Hyperparameters

We list below the hyperparameters used for all GANNO results, unless otherwise stated.

**Observations:**

- At a global level:
    - Train and test classification accuracy
    - Train and test classification loss
    - Boolean flag indicating if the loss is undefined or infinite
    - Training progress (current number of epochs/total epochs)
    - Ratio between the train and test loss (following Almeida et al. (2021))
    - Initial learning rate
    - Initial weight decay
- At a layerwise level:
    - Current learning rate
    - Previous action taken
    - Layer type (linear, convolutional, or attention)
    - Layer depth (an embedding which indicates if the current layer is first, intermediate, or final layer)
    - LAMB trust ratio (You et al., 2019) ($\frac{||\theta_l^{(t)}||}{||u_l^{(t)}||}$, where $\theta_l$ is the weights for layer $l$ and $u_l$ is the Adam update term)
    - Norm of gradients for the layer $||g_l^{(t)}||$
    - Norm of the updates $||u_l^{(t)}||$
    - Mean and variance of the weights $\theta_l^{(t)}$
    - Norm of the layer weights $||\theta_l^{(t)}||$

**Actions:**

Current learning rate ... $\{+0.00, \times 1.01, \times 1.10, \div 1.01, \div 1.10, +0.0005, -0.0005, +0.001, -0.001\}$

**PPO Details:**

- Number of executors/parallel copies of the environment = 4
- Max executor steps/number of training timesteps = 50 000
- Layer norm? = False
- Policy layer sizes = [128,128]
- Critic layer sizes = [64,64]
- Policy recurrent layer size = 64
- Policy layer size after recurrent layer = 64
- Epoch batch size = 32
- Sequence length = 8
- Number of epochs = 2
- Number of mini-batches = 4
- Normalise advantage? = True
- Normalise target values? = True
- Clip value? = True
- Normalise observations? = True

**Supervised learning of $f_\theta$:**

- Optimiser: Adam (Kingma and Ba, 2015)
- Weight decay: $\lambda = 0.1$

**Initial conditions:**

- Learning Rate, $\alpha_{\text{init}}$: Log-uniform distribution with bounds $[10^{-5}, 10^{-2}]$.
- Weight Decay, $\lambda$: Log-uniform distribution with bounds $[10^{-5}, 10^{-1}]$.

# B    Manual Schedules

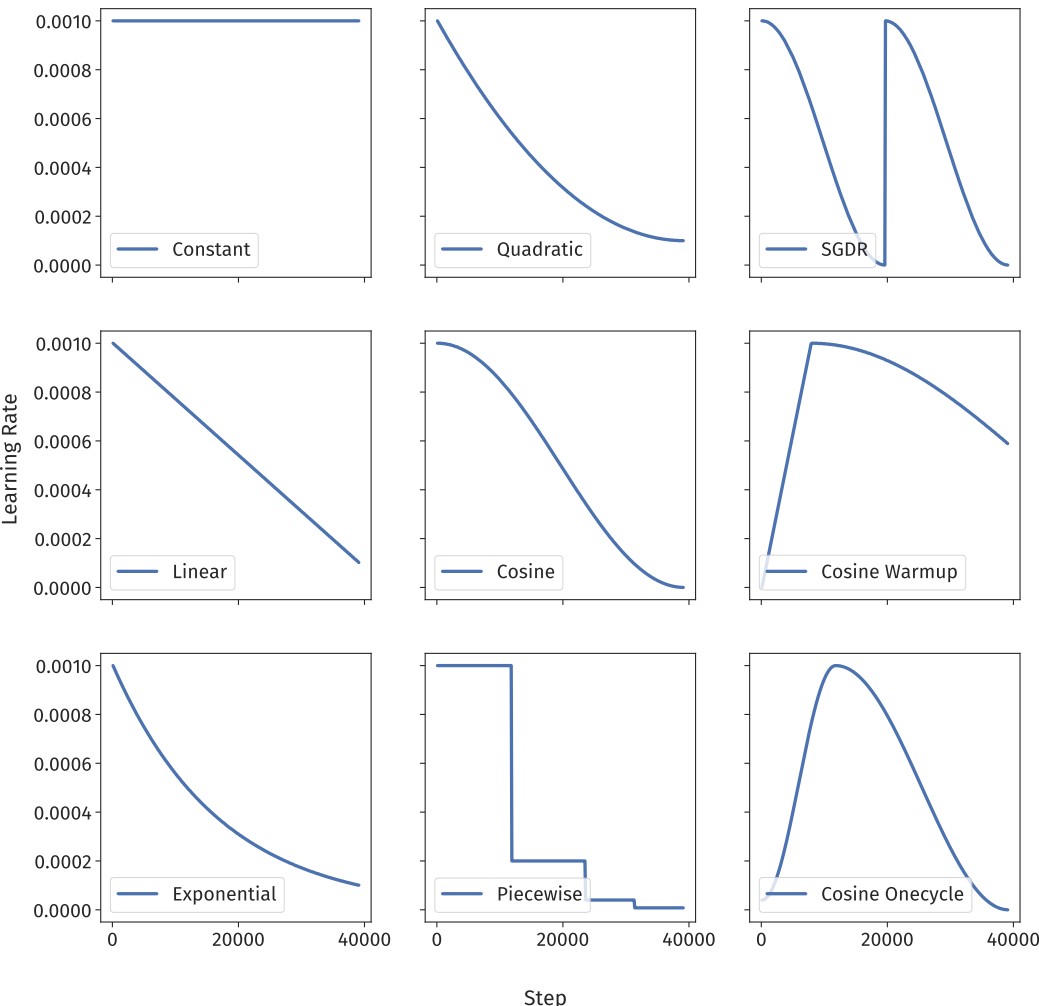

Figure 5: *Nine common handcrafted learning rate schedules used at various points in the paper.*

# C Extended Results

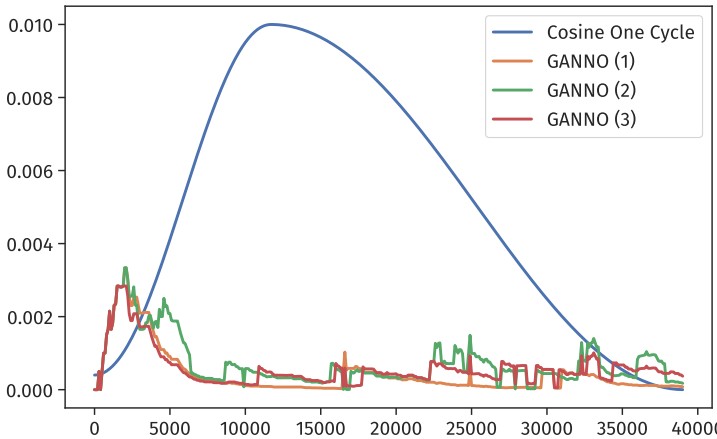

Figure 6: *Comparison of the manual learning rate schedule* `cosine-one-cycle` *with three instances of schedules from GANNO*. Notice how the GANNO agents act more conservatively in their scheduling to avoid the potential instability of a high learning rate.

Table 4: *Classification accuracies (%) achieved by GANNO, along with several simple learning rate schedules, using a five-layered CNN on* `CIFAR-10`*, across various initial learning rates, with* $\lambda = 0.1$. We see that GANNO performs competitively with the best manual schedules.

| | Initial Learning Rate | | | | | |
|---|---|---|---|---|---|---|
| **Simple schedules** | 0.0001 | 0.0003 | 0.001 | 0.003 | 0.01 | Average |
| Constant | 71.99 ± 0.38 | 71.33 ± 0.20 | 71.95 ± 0.43 | 73.25 ± 0.73 | 62.34 ± 0.45 | 70.17 ± 0.21 |
| Linear | 71.27 ± 0.58 | 72.51 ± 0.52 | 72.77 ± 0.13 | 73.19 ± 0.39 | 69.87 ± 0.72 | 71.92 ± 0.23 |
| Quadratic | 69.71 ± 0.50 | 73.12 ± 0.50 | 72.07 ± 0.83 | 72.63 ± 0.26 | 69.58 ± 1.42 | 71.42 ± 0.36 |
| Cosine | 70.85 ± 0.41 | 72.83 ± 0.35 | 73.26 ± 0.23 | 73.74 ± 0.69 | 69.82 ± 0.72 | 72.10 ± 0.23 |
| Exponential | 69.67 ± 0.59 | 72.99 ± 0.52 | 72.55 ± 0.24 | 72.47 ± 0.54 | 68.97 ± 0.74 | 71.33 ± 0.25 |
| Piecewise | 69.83 ± 0.44 | **73.56 ± 0.42** | 73.14 ± 0.24 | 73.42 ± 0.09 | 69.96 ± 0.71 | 71.98 ± 0.19 |
| SGDR | 70.72 ± 0.28 | 72.90 ± 0.36 | **73.79 ± 0.50** | **74.83 ± 0.09** | **71.36 ± 2.44** | **72.72 ± 0.51** |
| GANNO | **72.14 ± 0.77** | 73.44 ± 0.86 | 72.99 ± 1.37 | 73.08 ± 0.21 | 68.15 ± 2.35 | 71.96 ± 1.11 |

Table 5: *Classification accuracies (%) achieved by GANNO, along with several simple learning rate schedules, using a five-layered CNN on `CIFAR-10`, across various initial learning rates, with $\lambda = 0.0001$. We see that GANNO remains competitive with the other schedules, even at a lower weight decay value.*

| Simple schedules | Initial Learning Rate | | | | | Average |
|---|---|---|---|---|---|---|
| | 0.0001 | 0.0003 | 0.001 | 0.003 | 0.01 | |
| Constant | 70.72 ± 0.55 | 70.08 ± 0.81 | 69.95 ± 0.81 | 66.60 ± 0.26 | 57.88 ± 1.30 | 67.05 ± 0.37 |
| Linear | 70.43 ± 0.52 | 70.35 ± 0.37 | 72.33 ± 0.09 | 69.90 ± 0.79 | 59.67 ± 2.68 | 68.54 ± 0.57 |
| Quadratic | 69.67 ± 0.57 | 71.90 ± 0.95 | 71.97 ± 0.25 | 70.94 ± 0.74 | 59.40 ± 1.33 | 68.78 ± 0.38 |
| Cosine | 70.50 ± 0.38 | 70.70 ± 0.38 | 72.51 ± 0.22 | 70.29 ± 0.38 | 58.33 ± 2.24 | 68.47 ± 0.47 |
| Exponential | 69.33 ± 0.41 | 70.73 ± 0.34 | 72.10 ± 0.53 | 70.93 ± 0.14 | 57.98 ± 1.51 | 68.21 ± 0.34 |
| Piecewise | 69.85 ± 0.59 | 71.97 ± 0.44 | 72.08 ± 0.27 | 70.28 ± 0.54 | 60.04 ± 1.86 | 68.84 ± 0.42 |
| SGDR | 70.55 ± 0.41 | 70.91 ± 0.40 | **72.78 ± 0.43** | 70.54 ± 0.36 | 61.48 ± 1.48 | 69.25 ± 0.34 |
| VeLO | / | / | / | / | / | **74.86 ± 0.31** |
| LION | 71.49 ± 0.23 | **73.16 ± 0.27** | 36.02 ± 11.00 | 10.00 ± 0.00 | 10.00 ± 0.00 | 40.13 ± 2.30 |
| GANNO | **72.45 ± 0.32** | 71.14 ± 0.49 | 72.09 ± 0.65 | **71.08 ± 0.83** | **62.61 ± 1.25** | 69.87 ± 0.35 |

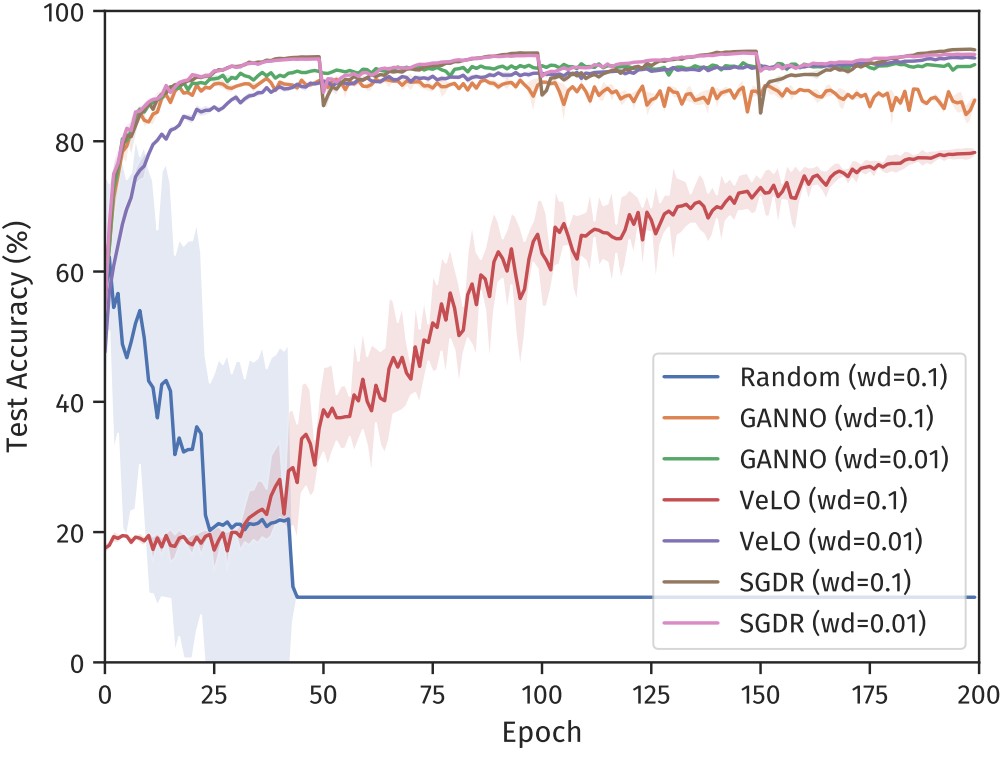

Figure 7: *Robustness of GANNO on ResNet-18.* Test accuracy across epochs for different schedules and learned optimisers on ResNet-18 trained on `CIFAR-10`, with an initial learning rate of $0.001$ for GANNO and the random agent. We see that GANNO produces robust and competitive schedules better able at handling different weight decay values. Note this is the same as 4 but it includes SGDR as an extra set of results.

