# OpenReview forum: "Generalisable Agents for Neural Network Optimisation"
_NeurIPS.cc/2023/Workshop/WANT — WANT@NeurIPS 2023 Poster_

### Official Review · Reviewer_rS9N · 2023-10-23
**In this review, we explore the paper "Generalisable Agents for Neural Network Optimization." The paper's writing and presentation are generally good, although there is room for improvement in notation. The author effectively conveys the motivation and purpose behind the work, addressing the critical issue of reducing human intervention in neural network training. The proposed method demonstrates success in reducing the need for human knowledge in the training process and uses separate datasets for training, testing, and validation to enhance generalization. The paper achieves promising results in standard cases but falls short in complex environments compared to other models. However, it excels in faster convergence, with a relatively small performance gap. Notably, the method is computationally efficient, making it environmentally friendly. In summary, the paper offers a generalizable approach to neural network optimization, making strides toward reduced human involvement, improved efficiency, and generalization.**

**Confidence:** 4

**Review:**

Introduction:
The paper titled "Generalizable Agents for Neural Network Optimization" presents a novel approach to optimizing neural networks with the goal of reducing the reliance on human knowledge and talent in the training process. The paper successfully conveys its motivation and purpose while demonstrating the potential of the proposed method. In this review, we will delve into the paper's strengths and areas for improvement, as well as its impact on the field of neural network optimization.

Writing and Presentation (Notation):
The writing and presentation of the paper are commendable, with clear explanations and logical flow. However, one aspect that could be improved is the notation for the summation operator. The paper's current notation could be potentially confusing, as it might lead readers to think that the index goes from t to T. A more unambiguous notation for summation (such as $\sum_{t=1}^{T}$) would enhance the paper's overall clarity and readability.

Motivation and Purpose:
The author effectively lays out the motivation and purpose of their work. The paper is driven by the idea of reducing the need for human intervention in the training of neural networks. This is a crucial issue in the field, and the author does an excellent job of highlighting the significance of addressing this challenge. The paper's clear articulation of its goals and objectives is a strong point.

Reduction of Human Involvement:
The paper succeeds in demonstrating how the proposed method reduces the reliance on human knowledge and talent for training neural networks. By introducing an agent-based approach, it offers a promising solution to automate and optimize the training process, which is an important step forward in the field of deep learning.

Data Separation:
The use of separate data sets for training, testing, and validation is a commendable practice, especially when striving for generalization. This ensures that the model's performance is not biased by overfitting to the training data and promotes more robust and adaptable solutions.

Performance on Vanilla Cases:
The paper reports promising results on standard or "vanilla" cases, indicating that the proposed method is effective in many situations. This is a positive outcome as it shows the potential of the approach in common neural network optimization scenarios.

Performance in Complex Environments:
However, it is noted that the proposed method does not outperform other models, particularly in more complex environments such as those involving ResNet architectures. It is essential to acknowledge this limitation and the fact that other handcrafted models still perform better in certain situations.

Convergence Speed:
One of the notable strengths of the proposed method is its ability to converge faster compared to other models. While it may not outperform more complex models, the gap in performance is relatively small. This attribute could be particularly valuable in scenarios where rapid model deployment is critical.

Computationally Efficient:
An important aspect highlighted in the paper is the computational efficiency of the proposed method. In today's world, where concerns about the environmental impact of deep learning are rising, a method that is computationally more efficient is a significant advantage. This factor can make the approach more attractive in real-world applications.

Generalizability:
In summary, the paper offers a generalizable method for neural network optimization, although it may not be the best-performing one across all scenarios. It represents a significant step towards reducing the reliance on human expertise in the training process, which aligns with the paper's motivating goal. The results demonstrate that the proposed method narrows the performance gap between it and other more complex models, particularly in vanilla cases. Additionally, the computational efficiency of the approach is a compelling feature in today's context.

Overall, the paper contributes to the ongoing effort to automate and generalize neural network optimization, and it has the potential to influence the field positively. While it may not be the definitive solution for all scenarios, it moves us closer to more automated, efficient, and generalizable methods for training neural networks.

---

### Official Review · Reviewer_EyqS · 2023-10-24
**Neural network optimization through an agent-layer interaction**

**Confidence:** 4

**Review:**

This paper proposes a new approach for neural network optimization through a MARL setting where each agent schedules hyperparameters of the neural network during training dynamically and responsively with respect to other agents' scheduling. More precisely, at each timestep, an agent interacts with a layer during $\tau$ ( $\approx$ 100) steps of training, the agent receives a global observation and a local observation of the current layer state; the global representation infers knowledge about the overall performance of the neural network and the local observation is specific to the agent performance (hyperparameter on which the agent acts).

I enjoyed reading the article, the approach seems interesting with nice experimental results and nice arguments in terms of computations and accuracy (it doesn't beat other competitors but still satisfies the trade-off in some sense). However, I am confused about some passages when authors talk about generalization, the generalization concerns the ability of agents to generalize to different neural architectures with different datasets in a zero-shot evaluation. I see the problem as context-dependent or data-dependent and generalization is more appropriate when there's a shift in the distribution of the same dataset but not a different dataset with different distribution.

I would also like to raise some points for the authors and also for the workshop organizers, a rigorous definition of the setting (not necessarily theoretic) is important when dealing with new approaches, it seems like there's a gap in the paper namely I would like to know how agents are trained, each agent doesn't interact with the same environment, for me it would make more sense to design the reward that includes this fact by taking into account the information flow between each environment (each layer), authors talked about global observation as an additional input to the agent and also a "no-action" for reward shaping, however, it's not explained how this is done in training. Maybe a setting where each agent observes a block of layers in a high-dimensional setting is suitable to the problem and relaxes this issue.
These comments don't aim to decrease the importance of this work but to criticize and understand a potentially interesting approach.

---

### Official Review · Reviewer_syen · 2023-10-24
**multi-agent RL to dynamically adjust the learning rate**

**Confidence:** 3

**Review:**

The authors propose to use multi-agent RL to dynamically adjust the learning rate at each layer of a neural net during training. The approach uses multi-agent RL with an agent at each layer of the network adjusts the learning rate. Previous has found faster convergence times with adaptive layerwise learning rates but these work required hyperparameters tuning.  In relation to RL, previous work used a single agent to evolve the learning rate for the entire network. The authors find that their proposed MARL setup, GANNO, can learn competitive schedules when compared to other leading approaches and is robust across a range of unseen initial conditions. The manuscript seems to fit nicely into the scope of the workshop.

The primary contribution is how to set up the MARL problem for adaptive layerwise learning rates. Generally, the paper is sufficient for the workshop. However, I should be improved as follows.

- A more detailed discussion of the improvement in computational resources, convergence times and/or hyper parameter search supported by comparisons with the alternative.
- More details on the experimental setup. How are the hyper-parameters of the RL selected; this could be a significant limitation. How robust are the agents to these choices. Likewise, how sensitive is the RL to the choice of time step size and action discretization.
- The networks and datasets study are relatively small. How well does the method scale to more realistic applications? Would a significant advantage emerge for VeLo on larger more complex settings?

---

### Official Review · Reviewer_Lf3t · 2023-10-25
**Review of "Generalisable Agents for Neural Network Optimisation"**

**Confidence:** 4

**Review:**

Summary:

Hyperparameters are vital for optimizing neural networks, and selecting them correctly has a substantial impact on optimization. GANNO is a multi-agent reinforcement learning approach designed to enhance neural network optimization. It achieves this by dynamically and responsively scheduling hyperparameters during training. GANNO employs an individual agent for each layer, which observes localized network dynamics and takes actions to adjust these dynamics on a per-layer basis. This collective effort at the layer level enhances overall performance.

Strengths:

The paper addresses the crucial topic of tuning hyperparameters in neural networks and attempts to introduce an automated and broadly applicable approach to achieve this objective.

Weaknesses:

The paper's claims about the performance of the proposed algorithm rely on a limited set of experiments, making it difficult to assess the significance of this work. Some of the reasoning and justifications presented in the paper lack cohesion. For example, the argument in Figure 3 is open to multiple interpretations and doesn't conclusively demonstrate that GANNO avoids local optima. It could simply indicate that the learning rate decreased too rapidly, impeding the learning process. Additionally, Table 1 illustrates that having an initial learning rate of 0.01 reduces accuracy, which contradicts the assertion that GANNO is unaffected by initial conditions.

Overall, The paper can be improved.

---

### Meta-Review · Area_Chair_iZJj · 2023-10-26

**Recommendation:** Accept (Poster)
**Confidence:** 4

**Metareview:**

The paper proposes an RL approach to tune layer-wise learning rates. Hyper-parameter tuning is computationally intensive, especially when it is as granular as the setting explored by the authors, and improving this process seems to align well with the scope of the workshop. Reviews are positive with the main concerns being lack of comparisons and some formalization details that are missing. The RL method does not outperform the state-of-the-art but reviewers found the approach sufficiently original and well explained.

---

### Decision · Program_Chairs · 2023-10-28

**Decision:**

Accept (Poster)

**Comment:**

We thank the authors for their time and contribution to WANT and we are pleased to share that after the reviewing process the paper has been accepted. Congratulations! We encourage the authors to consider reviewers' feedback for the improvement of the camera-ready version. We hope to see you in person at the workshop and brainstorm on efficient training research together!